# Comparison of Periodontal Status According to the Additives of Coffee: Evidence from Korean National Health and Nutrition Examination Survey (2013–2015)

**DOI:** 10.3390/ijerph16214219

**Published:** 2019-10-31

**Authors:** Yu-Rin Kim, Seoul-Hee Nam

**Affiliations:** 1Department of Dental Hygiene, Silla University, 140 Baegyang-daero, 700 beon-gil, Sasang-gu, Busan 46958, Korea; dbfls1712@hanmail.net; 2Department of Dental Hygiene, College of Health Science, Kangwon National University, 346 Hwangjo-gil, Dogye-up, Samcheok-si, Gangwon-do 25945, Korea

**Keywords:** coffee, dental care, national health insurance, periodontitis

## Abstract

It is well known that periodontal disease is highly related to dietary habits. As coffee is a typical beverage consumed worldwide, the relationship between coffee and periodontal disease was analyzed in this study using the data from the Korean National Health and Nutrition Survey (KNHANES) 2013–2015. Complex-samples chi square tests were performed for the comparison of the demographic characteristics of the 6528 study subjects and coffee components. Poisson linear regression analysis was performed for the analysis of the periodontal condition and coffee component effects, while complex-samples logistic regression analysis was performed to determine the demographic characteristics and coffee component effects. Over the years, the proportion of people drinking coffee with syrup or drinking a coffee mix containing both syrup and cream has decreased significantly. The results of the analysis, conducted by integrating the study subjects’ demographic characteristics and the coffee components, showed that the prevalence of periodontal disease was 0.83-times lower when drinking coffee with cream than when drinking black coffee. Coffee is the world’s second largest trade commodity following oil, and about 70%–80% of the world’s population drinks coffee. Drinking coffee with milk or cream can have a beneficial impact on periodontal disease.

## 1. Introduction

Periodontal disease is a chronic disease that is difficult to treat successfully. The alveolar bone is destroyed as the periodontal pocket is formed, which is the major cause of tooth loss [1]. The periodontal signs and progression of inflammation are associated with various factors, such as individual disposition, social factors, systemic factors, genetic factors, tooth condition, and microorganisms in biofilms [2]. Periodontal disease is an oral disease that is an important management target because it is closely related to many systemic diseases, such as diabetes [3], osteoporosis [4], and cardiovascular disease [5]. Fiorillo et al. [6] demonstrated that the release of bacterial products and inflammatory mediators into the bloodstream leads to systemic inflammation, and there are various etiological causes of this phenomenon. In particular, interferon gamma (IFN-γ) plays an important role in the progression of inflammation.

The aforementioned periodontal disease has adverse effects not only on the quality of life owing to physical, psychological, and social interactions, but also on food choices, thereby making it difficult for the person to acquire proper nutrition [7]. Periodontal disease is influenced by dietary patterns and food types. In particular, it is reported that dairy products and fruit consumption reduce the prevalence of periodontal disease [8]. Green tea extract [9] and vitamin [10] intake have also been associated with periodontal disease. The consumption of coffee, a typical beverage commonly consumed around the world, is rapidly increasing, and coffee currently ranks first among South Korea’s most frequently consumed food and beverage items. Adult men consume coffee 14 times a week, while women consume coffee 10 or more times a week [11]. The daily consumption has been found to range from as low as 140 mL to as high as 434 mL [12].

Coffee mix, which accounts for about 40% of the South Korean coffee market, is made in powder form by adding creamer and syrup to help alleviate the bitter, sour, and astringent tastes, and it is widely consumed due to its simplicity and universality [13]. Coffee creamer contributes to 15%–30% of the fat contained in coffee mix and contains vegetable palm oil as a main ingredient to maintain a consistent taste [14]. A study reported, however, that the intake of the fat components [15] used in coffee creamers and simple sugar adversely affects systemic and metabolic diseases [16]. Therefore, care should be taken when ingesting coffee creamer and sugar. 

Unlike in South Korea, in the West, coffee is mainly consumed by grinding and filtering coffee beans. Coffee beans contain chlorogenic acid, which functions as an antioxidant, and caffeine, for its excellent revitalizing effect. Studies on the prevention of Alzheimer’s disease, Parkinson’s disease, and oxidative stress in relation to coffee beans have also been conducted [17]. There have been many studies that have analyzed the relationship between coffee and related systemic diseases, but there is a dearth of research on the relationship between coffee and periodontal disease. Although there have been several studies examining the relationship between coffee intake amount and periodontal disease [18], there have been almost no studies analyzing the relationship between coffee intake and periodontal disease based on the coffee’s components.

Therefore, this study was conducted to assess the relationship between coffee’s components and periodontitis by using nationally representative data.

## 2. Materials and Methods

### 2.1. Study Subjects

This study used the data of the 6^th^ Korea National Health and Nutrition Examination Survey (KNHANES) conducted annually by the Centers for Disease Control and Prevention (CDC) since 2007. A total of 6528 people who responded to the survey items related to this study were selected. This study was conducted by obtaining the approval of the Research Ethics Review Committee of CDC for the 1st and 2nd years of the 6th round (2013-07CON-03-4C and 2013-12EXP-03-5C, respectively). After the 3rd year, it was overseen by the government for the public welfare and the survey was conducted without going through review by the Research Ethics Review Committee.

### 2.2. Demographic Characteristics

Based on the data from KNHANES, the gender, age, marriage, education, income, and economic activity status were examined. Age was classified into three categories: <20, ≥20; <40, ≥40; and <60, ≥60.

### 2.3. Characteristics of the Periodontal Condition

In terms of the periodontal condition characteristics of the subjects, the prevalence of periodontal disease was examined from the data obtained through oral examination, and the community periodontal index (CPI) was checked for the evaluation of the periodontal condition. For the CPI, the mouth was divided into sextants (maxillary right posterior, maxillary incisor, maxillary left posterior, mandibular right posterior, mandibular incisor, and mandibular left posterior), and the periodontal condition was measured using the ball probe designed by WHO. Code 0 refers to a healthy periodontal condition; Code 1 refers to the presence of gingival bleeding after probing; Code 2 refers to the presence of supragingival and subgingival calculus; Code 3 refers to the presence of a 5 mm periodontal pocket; and Code 4 refers to the presence of a ≥6mm pathological periodontal pocket. Greater severity of periodontal disease is indicated by a larger number. The periodontal disease measurement was performed by dentists [19].

### 2.4. Characteristics by Coffee Components

In the dietary questionnaire of the nutrition survey, the 2397 subjects who did not add both cream and syrup were classified into the Co (Coffee) group, the 39 subjects who added cream but did not add syrup were classified into the Co (Coffee) + Cr (Cream) group, the 264 subjects who did not add cream but added syrup were classified into the Co (Coffee) + Sy (Syrup) group, and the 3828 subjects who added both cream and syrup were classified into the Co (Coffee) + Cr (Cream) + Sy (Syrup) group.

### 2.5. Statistical Analyses

The data were analyzed using IBM SPSS ver. 21.0 (IBM Co., Armonk, NY, USA). For all the analyses, the stratification variable, clustering variable, and weight were applied. For the comparison of the coffee components according to the subjects’ demographic characteristics, complex-samples chi square tests were performed. As the unit of CPI is generally a positive integer, it is appropriate for the Poisson model. Therefore, Poisson linear regression analysis was performed to compare the effect on CPI. To analyze the independent effects of the subjects’ demographic characteristics and the coffee components on the prevalence of periodontal disease, complex-samples logistic regression analysis was performed. Model I was analyzed by focusing on the subjects’ demographic characteristics, model II was analyzed by focusing on the coffee components, and model III was analyzed by integrating the subjects’ demographic characteristics and the coffee components.

## 3. Results

### 3.1. Comparison of Coffee Components by Survey Year

Over the years, the proportion of people drinking black coffee has increased. Those who drank coffee more than three times a day drank at least five cups on average, consumed a coffee mix the most, and consumed coffee with cream the least (Table 1).

### 3.2. Demographic Characteristics by Coffee Component

Regardless of the gender and age, the proportion of people drinking a coffee mix containing both cream and syrup was high. Regardless of income and economic activity, the proportion of people drinking a coffee mix was higher. Employed people were more likely to drink a coffee mix whereas unemployed people were more likely to drink black coffee. There were significant differences only in age, marriage, and economic activity (Table 2) (p < 0.001).

### 3.3. CPI by Region and Coffee Component

The CPI of the maxillary right posterior region was 1.23-times higher when coffee with syrup was consumed and 1.09-times higher when a coffee mix was consumed, but was 0.88-times lower when coffee with cream was consumed compared to when black coffee was consumed. The CPI of the maxillary incisor region was 1.03-times higher when coffee with syrup was consumed than when black coffee was consumed and 1.18-times higher when a coffee mix was consumed, whereas it was 0.73-times lower when coffee with cream was consumed. The CPI of the maxillary left posterior region was 1.29-times higher when coffee with syrup was consumed, 1.09-times higher when a coffee mix was consumed, and 1.17-times higher when coffee with cream was consumed (Table 3). The CPI of the mandibular left posterior region was 1.36-times higher when coffee with syrup was consumed, 1.16-times higher when a coffee mix was consumed, and 0.76-times lower when coffee with cream was consumed compared to when black coffee was consumed. The CPI of the mandibular incisor was 1.11-times higher when coffee with syrup was consumed and 1.07-times higher when a coffee mix was consumed, but 0.74-times lower when coffee with cream was consumed. The CPI of the mandibular right posterior region was 1.45-times higher when coffee with syrup was consumed and 1.16-times higher when a coffee mix was consumed, but 0.51-times lower when coffee with cream was consumed (Table 4).

### 3.4. Effects of Demographic Characteristics and Coffee Components on the Prevalence of Periodontal Disease

With regard to model I, which was analyzed by focusing on the subjects’ demographic characteristics, the prevalence of periodontal disease was 1.87-times higher in males than in females. Regarding income, periodontal disease was 1.23-times higher in the middle-to-high-income group, 1.26-times higher in the middle-income group, 1.51-times higher in the middle-to-low-income group, and 1.68-times higher in the low-income group compared to the high-income group. There was a significant 1.07-times increase as age increased.

For model II, which was analyzed by focusing on the coffee components, the prevalence of periodontal disease was 1.03-times higher when coffee with only cream was consumed, 1.60-times higher when coffee with only syrup was consumed, and 1.09-times higher when a coffee mix containing both cream and syrup was consumed.

With regard to model III, which was analyzed by integrating the subjects’ demographic characteristics and the coffee components, the prevalence of periodontal disease was 1.92-times higher in males than in females. As for income, periodontal disease was 1.28-times higher in the middle-to-high-income group, 1.21-times higher in the middle-income group, 1.24-times higher in the middle-to-low-income group, and 1.58-times higher in the low-income group as compared to the high-income group. There was a significant 1.07-time increase as age increased. Periodontal disease prevalence was 0.83-times lower when cream was added, 1.60-times higher when syrup was added, and 1.16-times higher when a coffee mix containing both cream and syrup was consumed compared to drinking black coffee. The explanatory power of model III—analyzed by integrating the subjects’ demographic characteristics and the coffee components—was higher than that of model I—analyzed by focusing on the subjects’ demographic characteristics—and that of model II—analyzed by focusing on the coffee components(Table 5).

## 4. Discussion

Today, coffee is the second-largest world trade commodity following oil, and about 70%–80% of the world’s population drinks coffee [20]. Over the years, the proportion of people who drink black coffee with nothing added has increased significantly. As drinking coffee involves various social, cultural, political, and economic factors, it is necessary to investigate the type of coffee consumed according to the demographic characteristics. In the present study, the proportion of people who consume a coffee mix was high regardless of gender and age, which was consistent with the results of the study by Shin et al. [21] which also investigated coffee mix intake. Regardless of income and economic activity, the proportion of people who drink a coffee mix was high. The proportion of people who drink a coffee mix was higher among employed people than among unemployed people. This is in line with the results of the study by Shin et al. [21] which showed that 36.1% of the people who drink a coffee mix usually drink it at work, as coffee mix is always stocked at their workplace.

Meanwhile, 76% of the South Korean subjects habitually drink a coffee mix containing sugar and cream [16], which is known to cause dental caries [22]. A study has shown, however, that drinking a coffee mix reduces oral bacteria [23] because the phenolic compounds of coffee are strong antioxidants [24]. In particular, it has been reported that the cream added to the coffee in coffee mixes has recently been replaced with skimmed milk powder, thereby helping to prevent osteoporosis [25]. In the present study, aside from the maxillary left posterior region, the periodontal condition worsened when syrup was added or when a coffee mix was consumed over black coffee, but the condition improved when cream was added. As the daily consumption of black coffee will delay the alveolar bone regeneration process [26], drinking coffee with milk or cream can have a positive effect on the periodontal condition. This, however, does not consider the drinker’s demographic characteristics, but only the periodontal state and the coffee component, therefore there is a limit to the effect.

As coffee consumption increases, the prevalence of severe periodontal disease was shown to be 49.3% [18]. Therefore, people with progressed periodontal disease should drink coffee with care. Coffee consumption has been shown to be independently associated with an increased prevalence of tooth loss [27], and it has been suggested that coffee consumption has adverse effects on periodontal health [28]. In the present study, where model II was analyzed by focusing on the coffee components, the prevalence of periodontal disease was 1.60-times higher when only syrup was added, 1.09-times higher when a coffee mix containing both cream and syrup was consumed, and 1.03-times higher when only cream was added compared to when black coffee was consumed, thereby illustrating that adding cream to coffee appears to have no positive effect on the periodontal condition by region. In model III, however, the prevalence of periodontal disease was 0.83-times lower when cream was added than when black coffee was consumed without anything added. This result is consistent as the addition of cream to coffee has been shown to have a positive effect after adjusting for demographic characteristics. As the explanatory power of model III was higher than that of model I and II, the effect of cream can be regarded as positive. However, one report showed that cream has a negative influence on bone metabolism. Caffeine has been shown to increase bone loss and reduce bone healing after tooth extraction [29], and to inhibit the development of osteoblasts by decreasing the expression of vitamin D receptors on the surfaces of the osteoblasts [30]. Coffee whitener may also have a beneficial effect on periodontal disease. It has been suggested that the routine intake of lactic acid may have some direct effect on periodontal disease [31], but more studies are needed to determine the exact causal relationship between lactic acid and periodontal disease.

This study has several limitations. As it was a cross-sectional study that analyzed only the data from the 6th KNHANES, there is a limitation in explaining causality. As coffee intake was examined using a self-administered questionnaire, there is a limitation due to lack of objectivity of the subjects’ responses. As such, in future studies it will be necessary to conduct an in-depth analysis of the effect of coffee intake on the subjects’ overall health.

## 5. Conclusions

People with progressed periodontal disease should drink coffee with care. The consumption of coffee with cream may provide periodontal health benefits to South Korean adults. In addition, it may improve oral health by reducing the prevalence of periodontal disease.

## Figures and Tables

**Table 1 ijerph-16-04219-t001:** Comparison of Coffee Components by Survey Year (Unit: N [%]).

Year	Co	Co+Cr	Co+Sy	Co+Cr+Sy	*p*
2013	791 (33.3)	11 (0.4)	119 (5.1)	1474 (61.2)	0.000
2014	793 (37.0)	11 (0.7)	77 (3.5)	1266 (58.8)	
2015	813 (41.1)	17 (0.8)	68 (3.2)	1088 (54.9)	
Total	2397 (37.0)	39 (0.6)	264 (4.0)	3828 (58.4)	
†Daily coffee intake	5.34 ± 0.251	5.00 ± 0.000	5.35 ± 0.702	5.43 ± 0.107	

By chi square test using complex sampling design; Co (coffee), Cr (cream), and Sy (syrup). † by frequency analysis

**Table 2 ijerph-16-04219-t002:** Demographic Characteristics by Coffee Component (Unit: N [%]).

Characteristics		Co	Co+Cr	Co+Sy	Co+Cr+Sy	*p*
Gender	Male	1037 (37.6)	12 (0.5)	113 (3.8)	1661 (58.1)	0.472
	Female	1360 (36.4)	27 (0.8)	151 (4.1)	2167 (58.7)	
	Total	2397 (37.0)	39 (0.6)	264 (4.0)	3828 (58.4)	
Age	<20	550 (36.1)	4 (0.3)	47 (2.7)	925 (60.9)	0.029
	20–39	488 (36.9)	9 (0.8)	54 (4.2)	783 (58.2)	
	40–59	705 (38.8)	11 (0.4)	70 (3.8)	1066 (57.0)	
	60≤	654 (35.1)	15 (1.1)	93 (5.3)	1099 (58.6)	
	Total	2397 (37.0)	39 (0.6)	264 (4.0)	3828 (58.4)	
Marital Status	Married	1695 (32.0)	33 (0.7)	246 (4.5)	3364 (62.9)	0.000
	Single	699 (59.6)	6 (0.4)	18 (1.6)	464 (38.5)	
	Total	2394 (37.0)	39 (0.6)	264 (4.0)	3828 (58.4)	
Education	≥Elementary	878 (37.0)	12 (0.7)	99 (3.9)	1410 (58.4)	0.735
	Middle school	246 (35.2)	6 (0.5)	25 (2.9)	423 (61.3)	
	High school	589 (37.5)	9 (0.6)	66 (4.3)	910 (57.6)	
	College≤	529 (38.8)	11 (0.7)	57 (4.2)	794 (56.2)	
	Total	2242 (37.5)	38 (0.6)	247 (4.0)	3537 (57.9)	
Household Income	Low	480 (37.4)	6 (0.5)	52 (4.1)	705 (57.9)	0.936
	Middle-low	485 (37.1)	7 (0.5)	57 (4.5)	768 (57.9)	
	Middle	481 (37.4)	6 (0.5)	49 (3.8)	801 (58.3)	
	Middle-high	466 (35.7)	8 (0.5)	53 (3.8)	802 (60.0)	
	High	470 (37.3)	12 (1.0)	52 (3.6)	730 (58.1)	
	Total	2382 (36.9)	39 (0.6)	263 (4.0)	3806 (58.5)	
Economic Activity	Active	1208 (31.7)	18 (0.5)	165 (4.3)	2445 (63.6)	0.000
	None	909 (48.4)	16 (0.9)	64 (3.3)	894 (47.3)	
	Total	2117 (37.2)	34 (0.6)	229 (4.0)	3339 (58.3)	

By chi square test using complex sampling design; Co (coffee), Cr (cream), and Sy (syrup).

**Table 3 ijerph-16-04219-t003:** CPI of Maxillary by Region and Coffee Component.

Characteristics	Maxillary Right Posterior			Maxillary Incisor			Maxillary Left Posterior		
	*p*	OR	95% CI	*p*	OR	95% CI	*p*	OR	95% CI
Co+Cr+Sy	0.008	1.096	1.025–1.172	0.000	1.179	1.075–1.293	0.016	1.091	1.016–1.171
Co+Sy	0.010	1.225	1.051–1.428	0.783	1.033	0.820–1.302	0.002	1.293	1.103–1.516
Co+Cr	0.592	−0.881	0.553–1.402	0.383	−0.733	0.365–1.472	0.461	1.168	0.773–1.754
Co	·	1	·	·	1	·	·	1	·

By Poisson linear regression analysis.

**Table 4 ijerph-16-04219-t004:** CPI of Mandibular by Region and Coffee Component.

Characteristics	Mandibular Right Posterior			Mandibular Incisor			Mandibular Left Posterior		
	*p*	OR	95% CI	*p*	OR	95% CI	*p*	OR	95% CI
Co+Cr+Sy	0.000	1.163	1.078–1.253	0.018	1.065	1.011–1.122	0.000	1.164	1.085–1.250
Co+Sy	0.000	1.361	1.155–1.604	0.098	1.110	0.981–1.256	0.000	1.449	1.246–1.684
Co+Cr	0.320	−0.758	0.439–1.309	0.112	−0.735	0.503–1.075	0.035	−0.512	0.275–0.955
Co	·	1	·	·	1	·	·	1	·

By Poisson linear regression analysis.

**Table 5 ijerph-16-04219-t005:** Effects of the Subjects’ Demographic Characteristics and the Coffee Components on the Prevalence of Periodontal Disease.

Characteristics		Model I			Model II			Model III		
		*p*	OR	95% CI	*p*	OR	95% CI	*p*	OR	95% CI
Gender	Male	0.000	1.877	1.674–2.105				0.000	1.920	1.605–2.296
	Female	·	1	·				·	1	·
Marital Status	Married	0.850	1.014	0.876–1.174				0.897	−0.984	0.770–1.258
	Single	·	1	·				·	1	·
Education	<Elementary	0.093	-0.849	0.702–1.028				0.108	−0.794	0.599–1.052
	Middle school	0.600	1.053	0.867–1.279				0.530	−0.909	0.673–1.226
	High school	0.003	1.252	1.078–1.454				0.164	1.180	0.935–1.488
	>College	·	1	·				·	1	·
Household Income	Low	0.000	1.682	1.368–2.067				0.002	1.583	1.179–2.125
	Middle-low	0.000	1.509	1.244–1.831				0.155	1.243	0.921–1.679
	Middle	0.028	1.260	1.026–1.547				0.230	1.213	0.885–1.663
	Middle-high	0.026	1.227	1.025–1.469				0.070	1.279	0.980–1.669
	High	·	1	·				·	1	·
Economic Activity	Active	0.651	1.029	0.909–1.164				0.484	1.070	0.885–1.294
	None	·	1	·				·	1	·
Age		0.000	1.065	1.061–1.069				0.000	1.067	1.061–1.074
Co+Cr+Sy					0.263	1.093	0.450–0.864	0.114	1.162	0.965–1.400
Co+Sy					0.005	1.604	1.157–2.223	0.028	1.606	1.052–2.452
Co+Cr					0.943	1.027	0.489–2.160	0.654	−0.830	0.368–1.875
Co					·	1	·	·	1	·

By multivariable logistic regression analysis using complex sampling design; Co (coffee), Cr (cream), and Sy (syrup). Model I: Nagelkerke R square = 0.171, df1 = 11, df2 = 527, Wald F = 105.625, *p* < 0.001; Model II: Nagelkerke R square = 0.002, df1 = 3, df2 = 534, Wald F = 2.715, *p* < 0.05; Model III: Nagelkerke R square = 0.180, df1 = 14, df2 = 520, Wald F = 41.851, *p* < 0.001

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
