# Peer review of "Comparison of Periodontal Status According to the Additives of Coffee: Evidence from Korean National Health and Nutrition Examination Survey (2013–2015)"

_ijerph, 2019, doi:10.3390/ijerph16214219_

Round 1
Reviewer 1 Report
The manuscript is well structured and the topics fits within the Journal aim and scope.
Some minor concerns are about the periodontal disease and about the short slot of time(2013-2015) in which the study has been performed.
About the first issue, authors should increase the introduction section better defining the periodontal disease (etipatogeneisi and therapy) adding some more recent references
Fiorillo, L.; et al. Interferon Crevicular Fluid Profile and Correlation with Periodontal Disease and Wound Healing: A Systemic Review of Recent Data. Int. J. Mol. Sci. 2018, 19, 1908
Fiorillo, L. Chlorhexidine Gel Use in the Oral District: A Systematic Review. Gels 2019, 5, 31
About the second issue, it could be interesting if authors may able to increase the split time until 5 years break
overall an interesting paper.
Author Response
October 22, 2019
Dear Editor,
Please find enclosed our revised manuscript entitled “Association of Coffee Ingredient Affected by Periodontal Disease as a Risk Indicator: Evidence from the Korean National Health and Nutrition Examination Survey (2013-2015)” for the publication in the cluster issue of ‘International Journal of Environmental Research and Public Health’.
We are thankful to editor and referees for their careful and encouraging comments/ suggestions. Our answers to the comments are followed by a list of changes and are attached with this letter.
Sincerely yours,
Referee: 1
The manuscript is well structured and the topics fits within the Journal aim and scope.
Some minor concerns are about the periodontal disease and about the short slot of time(2013-2015) in which the study has been performed.
About the first issue, authors should increase the introduction section better defining the periodontal disease (etipatogeneisi and therapy) adding some more recent references
Fiorillo, L.; et al. Interferon Crevicular Fluid Profile and Correlation with Periodontal Disease and Wound Healing: A Systemic Review of Recent Data. Int. J. Mol. Sci. 2018, 19, 1908
Fiorillo, L. Chlorhexidine Gel Use in the Oral District: A Systematic Review. Gels 2019, 5, 31
About the second issue, it could be interesting if authors may able to increase the split time until 5 years break
overall an interesting paper.
Answer
We have described the etiological content in the introduction.In your opinion, the next article will be dated and analyzed extensively. Therefore, this part has been presented as a limitation in consideration.

Reviewer 2 Report
Review report: ijerph-623102 Association of Coffee Ingredient Affected by Periodontal Disease as a Risk Indicator: Evidence from the Korean National Health and Nutrition Examination Survey (2013-2015)
This is an analysis of a national helath and nutrition survey conducted in Korea between 2013 to 2015. The authors aimed to investigate whether there was any associations between coffee drinking (with and without cream or syrup) and the severity of periodontal disease in all age groups.
I have major concerns with regard to the hypothesis of this study and the methodology used to analyse the data.
The title of the paper is confusing and needs to be reconsidered. The authors do not present a convincing literature review to provide evidence of any biological plausibility underpinning their hypothesis. They make claims of dietary impacts on the periodontium based on single cross-sectional surveys. They need to be more specific about the aetiological factors that affect periodontal health and provide explanations of the biological pathways by which these factors have these effects. The Methods are not clearly explained, especially in relation to: The outcome: What do the range of CPI measures mean and how did they classified periodontal disease in this study. How was CPI measured and by whom? Did all participants have CPI measures recorded? What were the descriptive statistics of CPI? My strongest concerns were with the statistical analyses – despite a broad range of demographic and behaviour variables being obtained during the survey the authors chose to not include confounders in their regression modelling (linear and logistic regression). The reported unadjusted univariate analyses but not multivariate analyses – I do not understand why they did not undertake these adjusted multivariate analyses when they had the data to do this. Essential confounders in these analyses would be, at the very least, age, sex, smoking status, socioeconomic status. As such, the Results are limited in their meaning and so I suggest that their discussion and conclusion are not supported by sound statistical modelling.Author Response
October 22, 2019
Dear Editor,
Please find enclosed our revised manuscript entitled “Association of Coffee Ingredient Affected by Periodontal Disease as a Risk Indicator: Evidence from the Korean National Health and Nutrition Examination Survey (2013-2015)” for the publication in the cluster issue of ‘International Journal of Environmental Research and Public Health’.
We are thankful to editor and referees for their careful and encouraging comments/ suggestions. Our answers to the comments are followed by a list of changes and are attached with this letter.
Sincerely yours,
Referee: 2
Review report: ijerph-623102 Association of Coffee Ingredient Affected by Periodontal Disease as a Risk Indicator: Evidence from the Korean National Health and Nutrition Examination Survey (2013-2015)
This is an analysis of a national helath and nutrition survey conducted in Korea between 2013 to 2015. The authors aimed to investigate whether there was any associations between coffee drinking (with and without cream or syrup) and the severity of periodontal disease in all age groups.
I have major concerns with regard to the hypothesis of this study and the methodology used to analyse the data.
The title of the paper is confusing and needs to be reconsidered. The authors do not present a convincing literature review to provide evidence of any biological plausibility underpinning their hypothesis. They make claims of dietary impacts on the periodontium based on single cross-sectional surveys. They need to be more specific about the aetiological factors that affect periodontal health and provide explanations of the biological pathways by which these factors have these effects. The Methods are not clearly explained, especially in relation to: The outcome: What do the range of CPI measures mean and how did they classified periodontal disease in this study. How was CPI measured and by whom? Did all participants have CPI measures recorded? What were the descriptive statistics of CPI? My strongest concerns were with the statistical analyses – despite a broad range of demographic and behaviour variables being obtained during the survey the authors chose to not include confounders in their regression modelling (linear and logistic regression). The reported unadjusted univariate analyses but not multivariate analyses – I do not understand why they did not undertake these adjusted multivariate analyses when they had the data to do this. Essential confounders in these analyses would be, at the very least, age, sex, smoking status, socioeconomic status. As such, the Results are limited in their meaning and so I suggest that their discussion and conclusion are not supported by sound statistical modelling.
Answer
We modified the title.
The introduction is complemented with etiological factors that affect periodontal health.
The CPI has been supplemented in detail.
Table 3 analyzes the periodontal state of each part according to the coffee ingredients. Therefore, demographic characteristics were not considered, and this was presented as a limit point in the consideration. However, Table 4 covers Table 3. Therefore, Table 4 presents the revised demographics. Table 4 presents the adjusted demographics characteristics.)
In Table 4, logistic regression analysis was performed to examine the demographic characteristics and the effects of coffee on the prevalence of periodontal disease. Model 1 examined the effect of periodontal disease prevalence according to demographic characteristics. Model 1 examined the effect of periodontal disease prevalence according to Additives of Coffee. Model 3 identifies the effects of coffee additives on the prevalence of periodontal disease, taking into account demographic characteristics (gender, age, education, marital status, income, etc.). Therefore, the limitations in Table 3 have been added to the consideration. and The description in Table 4 has been added for the reader to understand.)

Reviewer 3 Report
In this manuscript, the authors examined the relationship between coffee and periodontal disease using Korean National Health and Nutrition Survey based on coffee intake and the components of coffee. The study is based on the survey of 6.528 people who responded to the survey.
Overall, the manuscript is easy to follow. Unfortunately, a lot of cited references are dated. I think, the manuscript should be enriched with additional references. More than 30% references are older than 10 years.
The title of the paper has to be rewritten. There is no risk assessment performed and the current title indicates that the publication covers this topic.
The employed methods are appropriate and the authors made a huge effort to evaluate such a big group of participants. The results are presented clearly. However, the conclusions should be presented in a broader context. The weakest part of the paper is the conclusion section! The results are the statements. The manuscript provides a snapshot of periodontal disease in Korea. There is no explanation of the importance or relevance of the study. I recommend to present the results in a broader, international perspective.
Author Response
October 22, 2019
Dear Editor,
Please find enclosed our revised manuscript entitled “Association of Coffee Ingredient Affected by Periodontal Disease as a Risk Indicator: Evidence from the Korean National Health and Nutrition Examination Survey (2013-2015)” for the publication in the cluster issue of ‘International Journal of Environmental Research and Public Health’.
We are thankful to editor and referees for their careful and encouraging comments/ suggestions. Our answers to the comments are followed by a list of changes and are attached with this letter.
Sincerely yours,
Referee: 3
In this manuscript, the authors examined the relationship between coffee and periodontal disease using Korean National Health and Nutrition Survey based on coffee intake and the components of coffee. The study is based on the survey of 6.528 people who responded to the survey.
Overall, the manuscript is easy to follow. Unfortunately, a lot of cited references are dated. I think, the manuscript should be enriched with additional references. More than 30% references are older than 10 years.
The title of the paper has to be rewritten. There is no risk assessment performed and the current title indicates that the publication covers this topic.
The employed methods are appropriate and the authors made a huge effort to evaluate such a big group of participants. The results are presented clearly. However, the conclusions should be presented in a broader context. The weakest part of the paper is the conclusion section! The results are the statements. The manuscript provides a snapshot of periodontal disease in Korea. There is no explanation of the importance or relevance of the study. I recommend to present the results in a broader, international perspective.
Answer
We recently changed to references.
We modified the title.
We have revised our conclusions to the broader and global side.

Round 2
Reviewer 2 Report
This paper is vastly improved from the previous version and I commend the authors for considering the comments and addressing the concerns raised.
I consider the revisions to be acceptable and recommend this paper be accepted by the journal. I hope the authors will build on this preliminary work by undertaking more comprehensive and complex statistical analyses (controlling for other common confounders) and exploring the biological pathways underpinning these potentially interesting findings to strengthen the level of evidence reported.
There are still some minor edits required after the tracked changes have been accepted.